# Preventing Collapse in Contrastive Learning with Orthonormal Prototypes (CLOP)

## Abstract

Contrastive learning has emerged as a powerful method in deep learning, excelling at learning effective representations through contrasting samples from different distributions. However, dimensional collapse, where embeddings converge into a lower-dimensional space, poses a significant challenge, especially in semi-supervised and self-supervised setups. In this paper, we first theoretically analyze the effect of large learning rates on contrastive losses that solely rely on the cosine similarity metric, and derive a theoretical bound to mitigate this collapse. Building on these insights, we propose **CLOP**, a novel semi-supervised loss function designed to prevent dimensional collapse by promoting the formation of orthogonal linear subspaces among class embeddings. Unlike prior approaches that enforce a simplex ETF structure, CLOP focuses on subspace separation, leading to more distinguishable embeddings. Through extensive experiments on real and synthetic datasets, we demonstrate that CLOP enhances performance, providing greater stability across different learning rates and batch sizes.

## 1 Introduction

Recent advancements in deep learning have positioned **Contrastive Learning** as a leading paradigm, largely due to its effectiveness in learning representations by contrasting samples from different distributions while aligning those from the same distribution. Prominent models in this domain include SimCLR Chen et al. (2020a), Contrastive Multiview Coding (CMC) Tian et al. (2020a), VICReg Bardes et al. (2021), BarLowTwins Zbontar et al. (2021), among others Wu et al. (2018); Henaff (2020); Li et al. (2020). These models share a common two-stage framework: representation learning and fine-tuning. In the first stage, representation learning is performed in a self-supervised manner, where the model is trained to map inputs to embeddings using contrastive loss to separate samples from different labels. In the second stage, fine-tuning occurs under a supervised setup, where labeled data is used to classify embeddings correctly. For practical applicability, a small amount of labeled data is required in the fine-tuning stage to produce meaningful classifications, making the overall pipeline semi-supervised.

Empirical evidence demonstrates that these models, even with limited labeled data (as low as 10%), can achieve performance comparable to fully-supervised approaches on moderate to large datasets Jaiswal et al. (2020).

Despite the effectiveness of contrastive learning on largely unlabeled datasets, a common issue encountered during the training process is **Dimensional Collapse**. As pointed out by Jing et al. (2021); Fu et al. (2022); Rusak et al. (2022); Xue et al. (2023); Gill et al. (2024); Tao et al. (2024); Hassanpour et al. (2024), this

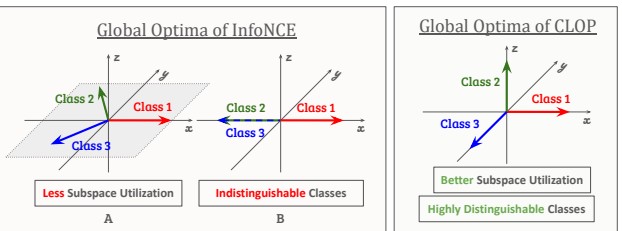

Figure 1: Illustration of global optima for InfoNCE and **CLOP** (this paper). For InfoNCE, global optima are reached when the model merges samples of the same class into a single embedding, whether the class arrangement is ETF (A) or co-linear (B). In contrast, the proposed CLOP introduces a novel regularizer that encourages embeddings to occupy a highly separable, full-rank space.

phenomenon describes the collapse of output embeddings from the neural network into a lower-dimensional space, reducing their spatial utility and leading to indistinguishable classes (see Figure 1.B). There are two main approaches to resolve this issue: augmentation modification Jing et al. (2021); Xue et al. (2023); Fu et al. (2022); Tao et al. (2024) and loss modification Fu et al. (2022); Rusak et al. (2022); Hassanpour et al. (2024). In this paper, we propose an additional term in the loss function to address the issue of collapse. Our approach to deal with Dimensional collapse involves selecting prototypes similarly to Zhu et al. (2022); Gill et al. (2024). The key distinction lies in the fact that, while their approach enforces the embeddings to conform to a simplex Equiangular Tight Frame (ETF) hyperplane, our method aims to push the embeddings toward distinct orthogonal linear subspaces, allowing them to occupy the full-rank space (see Figure 1 for intuition). This result in more distinguishable subspace clusters, which can be more effectively learned by the downstream classifier.

The **main contributions of this paper** can be summarized in three perspectives. First, we theoretically identify the impact of an overly large learning rate on contrastive learning loss that is based solely on cosine similarity as the metric. We provide a theoretical bound for the learning rate to avoid collapse for $k$ classes under specified conditions. Furthermore, we simplify this bound to a constant $O(1)$. Second, we analyze the results under moderate learning rates and observe that the embeddings naturally lie on a hyperplane, which reduces spatial usage and makes it more difficult for the downstream classifier to learn effectively. Finally, with these findings, we propose a novel loss term, **CLOP**, which involves pulling a partial training dataset towards a few orthonormal prototypes. This loss is applicable in both semi-supervised and fully-supervised contrastive learning settings, where a subset of labeled data is available for training. Through extensive experiments, we demonstrate the performance superiority of CLOP. Specifically, we show that CLOP is significantly more stable across different learning rates and smaller batch sizes.

**Paper Organization** In Section 2, we begin by discussing the necessary background, including recent advancements in both self-supervised and supervised contrastive learning, as well as the analysis of the Dimensional collapse phenomenon in deep learning and contrastive learning, in particular. In Section 3, we present our theoretical analysis of Dimensional collapse. Next, in Section 4, we introduce our proposed model, **CLOP**. Lastly, we present the experimental results on CIFAR-100 and Tiny-ImageNet in Section 5.

## 2 RELATED WORK

Contrastive learning has gained prominence in deep learning for its ability to learn meaningful representations by pulling together similar (positive) pairs and pushing apart dissimilar (negative) pairs in the embedding space. Positive pairs are generated through techniques like data augmentation, while negative pairs come from unrelated samples, making contrastive learning particularly effective in self-supervised tasks like image classification. Pioneering models such as SimCLR Chen et al. (2020a), CMC Tian et al. (2020a), VICReg Bardes et al. (2021), and Barlow Twins Zbontar et al. (2021) share the objective of minimizing distances between augmented versions of the same input (positive pairs) and maximizing distances between unrelated inputs (negative pairs). SimCLR maximizes agreement between augmentations using contrastive loss, while CMC extends this to multi-view learning Chen et al. (2020a); Tian et al. (2020a). VICReg introduces variance-invariance-covariance regularization without relying on negative samples Bardes et al. (2021), and Barlow Twins reduce redundancy between different augmentations Zbontar et al. (2021).

Recent innovations have improved contrastive learning across various domains. For instance, methods like structure-preserving quality enhancement in CBCT images Kang et al. (2023) and false negative cancellation Huynh et al. (2022) have enhanced image quality and classification accuracy. In video representation, cross-video cycle-consistency and inter-intra contrastive frameworks Wu & Wang (2021); Tao et al. (2022) have shown significant gains. Additionally, contrastive learning has advanced sentiment analysis Xu & Wang (2023), recommendation systems Yang et al. (2022), and molecular learning with faulty negative mitigation Wang et al. (2022b). Xiao et al. (2024) introduces GraphACL, a novel framework for contrastive learning on graphs that captures both homophilic and heterophilic structures without relying on augmentations.

## 2.1 CONTRASTIVE LOSS

In unsupervised learning, Wu et al. (2018) introduced InfoNCE, a loss function defined as:

$$\mathcal{L}_{\text{infoNCE}} = -\sum_{i \in I} \log \frac{\exp(\mathbf{z}_i^\top \mathbf{z}_{j(i)}/\tau)}{\sum_{a \neq i} \exp(\mathbf{z}_i^\top \mathbf{z}_a/\tau)} \tag{1}$$

where $\mathbf{z}_i$ is the embedding of sample $i$, $j(i)$ its positive pair, and $\tau$ controls the temperature.

Recent refinements focus on (1) component modifications, (2) similarity adjustments, and (3) novel approaches. Li et al. (2020) use EM with k-means to update centroids and reduce mutual information loss, while Wang et al. (2022a) add L2 distance to InfoNCE, though both underperform state-of-the-art (SOTA) techniques. Xiao et al. (2020) reduce noise with augmentations, and Yeh et al. (2022) improve gradient efficiency with Decoupled Contrastive Learning, though neither surpasses SOTA. In similarity adjustments, Chuang et al. (2020) propose a debiased loss, and Ge et al. (2023) use hyperbolic embeddings, but neither outperforms SOTA. Novel methods include min-max InfoNCE Tian et al. (2020b), Euclidean-based losses Bardes et al. (2021), and dimension-wise cosine similarity Zbontar et al. (2021), achieving competitive performance without softmax-crossentropy.

## 2.2 SEMI-SUPERVISED AND SUPERVISED CONTRASTIVE LEARNING

Semi-supervised contrastive learning effectively leverages both labeled and unlabeled data to learn meaningful representations. Zhang et al. (2022) introduced a framework with similarity co-calibration to mitigate noisy labels by adjusting the similarity between pairs. Inoue & Goto (2020) proposed a Generalized Contrastive Loss (GCL), unifying supervised and unsupervised learning for speaker recognition, while Kim et al. (2021) combined contrastive self-supervision with consistency regularization in SelfMatch. In domain adaptation, Singh (2021) utilized class-wise and instance-level contrastive learning to minimize domain gaps, while Liu & Abdelzaher (2021) developed a method for Human Activity Recognition (HAR) using semi-supervised contrastive learning to achieve state-of-the-art performance. In medical image segmentation, Hua et al. (2022) introduced uncertainty-guided voxel-level contrastive learning, and Hu et al. (2021) combined global self-supervised and local supervised contrast for improved label efficiency. For automatic speech recognition (ASR), Xiao et al. (2021) reduced reliance on large labeled datasets while maintaining high accuracy using semi-supervised contrastive learning. In 3D point cloud segmentation, Jiang et al. (2021) presented a pseudo-label contrastive framework, while Shen et al. (2021) employed contrastive learning for intent discovery in conversational datasets with minimal labeled data.

Supervised contrastive learning, initially proposed by Khosla et al. (2020), extends contrastive loss to fully supervised settings, significantly improving task performance. Graf et al. (2021) further explored its relationship with cross-entropy loss, highlighting its advantages in feature learning, while Cui et al. (2021) introduced learnable class centers to balance class representations. Domain-specific applications include recommendation systems, where supervised contrastive learning enhances item representations Yang et al. (2022), and product matching, where it improves matching accuracy Peeters & Bizer (2022). It has also been extended to natural language processing as a fine-tuning objective for pre-trained models Gunel et al. (2020). To address imbalanced datasets and noisy labels, Targeted Supervised Contrastive Learning (TSC) focuses on under-represented classes in long-tailed recognition Li et al. (2022b), while Selective-Supervised Contrastive Learning (Sel-CL) selectively learns from clean data to improve performance under noisy supervision Li et al. (2022a).

## 2.3 DIMENSIONAL COLLAPSE

Dimensional collapse is a notable phenomenon in deep learning, particularly during the terminal phase of training. Several works have focused on establishing a theoretical foundation for understanding dimensional collapse through geometric and optimization properties. Zhu et al. (2021) provide a geometric framework that highlights the alignment of classifiers and features in neural networks with a Simplex ETF structure. Similarly, Mixon et al. (2022) explore it from the perspective of unconstrained features, showing that collapse naturally occurs without explicit regularization. Ji et al. (2021) extend this with an unconstrained layer-peeled model, linking collapse to optimization processes. Yaras et al. (2022) use Riemannian geometry to show that collapse solutions are global minimizers. Extensions of dimensional collapse to more complex settings include Jiang et al.

(2023), who broaden its study to networks with a large number of classes. Rangamani et al. (2023) analyze intermediate phases of collapse, while Tirer et al. (2023) show that practical networks rarely achieve exact collapse, yet approximate collapse still occurs. Galanti et al. (2021) explore its role in transfer learning, demonstrating improvements in generalization. Zhong et al. (2023) apply dimensional collapse to imbalanced semantic segmentation, highlighting its impact on class separation and feature alignment.

For dimensional collapse in contrastive learning, Jing et al. (2021) examine dimensional collapse in self-supervised learning. They attribute this to strong augmentations distorting features and implicit regularization driving weights toward low-rank solutions. To address this, they propose DirectCLR, which optimizes the representation space and outperforms SimCLR on ImageNet by better preventing collapse. Similarly, Xue et al. (2023) explore how simplicity bias leads to class collapse and feature suppression, with models favoring simpler patterns over complex ones. They suggest increasing embedding dimensionality and designing augmentation techniques that preserve class-relevant features to counter this bias and promote diverse feature learning. Similarly, Fu et al. (2022) emphasize the role of data augmentation and loss design in preventing class collapse, proposing a class-conditional InfoNCE loss term that uniformly pulls apart individual points within the same class to enhance class separation. In supervised contrastive learning, Gill et al. (2024) propose loss function modifications to follow an ETF geometry by selecting prototypes that form this structure. In graph contrastive learning, Tao et al. (2024) introduce a whitening transformation to decorrelate feature dimensions, avoiding collapse and enhancing representation capacity. In medical image segmentation, Hassanpour et al. (2024) address dimensional collapse through feature normalization and whitening approach to preserve feature diversity. Finally, Rusak et al. (2022) investigate the preference of contrastive learning for content over style features, leading to collapse. They propose to leverage adaptive temperature factors in the loss function to improve feature representation quality.

## 3 THEORETICAL ANALYSIS

The first part of this section delves into a theoretical examination of *complete collapse*, which refers to the phenomenon where all embeddings converge to a single point in contrastive learning. Initially, we show that complete collapse is a local minimum for the InfoNCE loss by showing that linear embeddings result in a zero gradient (Lemma 1). This result is demonstrated using the InfoNCE loss, but it can be applied to the majority of current loss functions that rely solely on cosine similarity as a similarity metric. This holds across various settings, including unsupervised Henaff (2020); Chen et al. (2020a); Cui et al. (2021); Xiao et al. (2020); Yeh et al. (2022); Wang et al. (2022a), semi-supervised Hu et al. (2021); Shen et al. (2021), and supervised contrastive learning Khosla et al. (2020); Cui et al. (2021); Peeters & Bizer (2022); Li et al. (2022b). Subsequently, we discuss the effect of a large learning rate in producing complete collapse. We derive an upper-bound on the learning rate to avoid collapse under mild assumptions (Theorem 1).

In the latter part of this section, we examine the phenomenon of *dimensional collapse*, where the embedding space of a model progressively shrinks to a lower-dimensional subspace (Lemma 2). We describe how cosine-similarity optimization drives this collapse without achieving the formation of a Simplex Equiangular Tight Frame (ETF). Specifically, we demonstrate that under gradient descent, which minimizes the total class cosine similarity loss $\mathcal{L}_{\text{class}}$, the embeddings — initially spanning the full space — contract into a lower-dimensional subspace where they are not equidistant (not ETF).

### 3.1 LARGE LEARNING RATE CAUSES COMPLETE COLLAPSE

The concept of the InfoNCE loss, as defined in Eq. (1), aims to encourage the embeddings to form distinguishable clusters in high-dimensional space, thereby facilitating classification for downstream models. However, in Lemma 1, we demonstrate that the worst-case scenario — where all embeddings become identical or co-linear — also constitutes a local optimum for the InfoNCE loss. This includes non-unique global optima, which we will further explore in Section 3.2. This observation suggests that, from a theoretical perspective, InfoNCE exhibits instability, as both the best and worst solutions can lead to stationary points.

**Lemma 1.** *Let $\mathcal{F} : \mathbb{R}^m \to \mathbb{R}^{m'}$ be a family of Contrastive Learning structures, where $m$ and $m'$ denote the dimensions of the inputs and embeddings, respectively. If a function $f \in \mathcal{F}$ is trained using the InfoNCE loss, then there exist infinitely many local minima where all embeddings produced by $f$ are all equal or co-linear.*

The proof of Lemma 1 relies on the observation that the embeddings are only compared against each other. If all embeddings are either identical or co-linear, the gradient vanishes due to the lack of angular differences, as well as the normalization process. The full proof of Lemma 1 is presented in Appendix C.

The remainder of this section focuses on the causes of reaching local minima. In particular, we examine the role of a large learning rate in causing complete collapse. To understand the dynamics of contrastive learning, it is crucial to consider two forces acting on each embedding: the *gravitational force* within the same class and the *repulsive force* between different classes. Contrary to the common belief that the gravitational force is responsible for inducing collapse, we observe that the primary cause of complete collapse in contrastive learning is the overshooting of the repulsive force.

Figure 2 illustrates this phenomenon. For simplicity of analysis, assume that the model has successfully merged samples from the same class into a single *class embedding*. Each light blue dot represents one class embedding, while the dark blue dot represents the mean of all class embeddings. In practice, the mean is unlikely to be located at the center of the space. The purpose of the repulsive force is to arrange the class embeddings more uniformly across the space and center the mean of the class embeddings, as shown in Figure 2.B with a small learning rate.

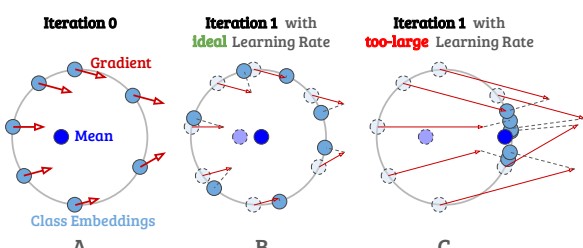

Figure 2: Illustration of the effect of repulsive force in contrastive learning. Light blue dots represent the individual class embedding, while the dark blue dot represents the mean of all class embeddings.

However, when the learning rate is too high, the repulsive force causes the class embeddings to overshoot, as shown in Figure 2.C with a large learning rate. Due to the normalization operation inherent in contrastive learning, the class embeddings tend to converge, which ultimately results in the local minima characterized as complete collapse, as stated in Lemma 1. This overshooting phenomenon necessitates the upper-bounding of the learning rate to better regulate the repulsive force, ensuring optimal space utilization of class embeddings.

In Theorem 1, we establish an upper bound for the learning rate of the gradient descent step, where $k$ represents the total number of class embeddings. This result is derived under the same assumption that embeddings of the same class are aggregated into a single class embedding by the upstream model. As shown in Figure 2, the shift in the class embedding mean is a critical phenomenon associated with complete collapse. Consequently, the upper bound is obtained by constraining the movement of the class embedding mean, ensuring that the mean in the next step remains strictly non-increasing.

**Theorem 1.** *Consider $k$ class embeddings uniformly distributed on the surface of an $m$-dimensional unit ball, where $m \geq k > 2$. The upper bound on the learning rate $\mu$ for the gradient descent step, to minimize cosine similarity scores between class embeddings while preventing the class embedding mean from increasing by a ratio of $(1 + \varepsilon)$, is given by:*

$$(1 - \eta)^2 \leq \left(1 + \frac{\eta}{k - 1} - \frac{2\eta}{k} - 2\frac{\eta^2}{k(k - 1)} + \frac{\eta^2 k}{(k - 1)^2}\right)(1 + \varepsilon)^2. \qquad (2)$$

*Setting $\varepsilon = 0$ guarantees a non-increasing mean, and provides an $O(1)$ upper bound for the learning rate.*

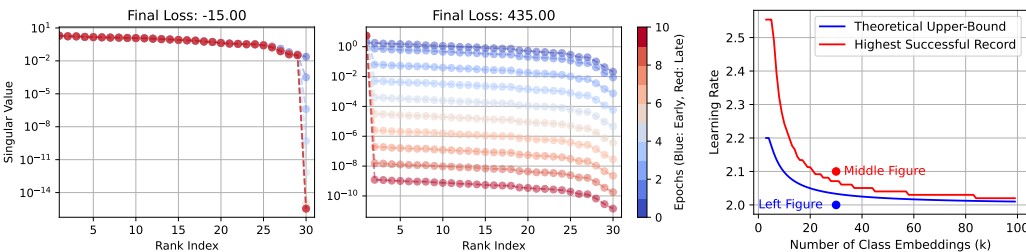

Figure 3: Numerical experiment conducted on tightness of Theorem 1. **Left & Middle:** Singular value spectra of $\mathbf{X}$ at different training epochs (color-coded from blue to red). The **Left** panel shows successful optimization with a learning rate of 2.0, while the **Middle** panel demonstrates optimization failure (complete collapse) at a learning rate of 2.1. **Right:** The maximum learning rates preventing collapse over 5 consecutive trials, for varying class embedding sizes, are plotted against the theoretical upper bound ($\varepsilon = 0$) from Theorem 1.

The proof proceeds by establishing an upper bound on the norm of each class embedding, which is dependent on the step size after a gradient descent step. This enables us to derive an upper bound for the shifted mean. By comparing the upper bound of the shifted mean with the original mean, we guarantee that the mean remains non-increasing throughout the gradient descent process, thus preventing complete collapse. The detailed proof of Theorem 1 can be found in Appendix D.

To study the tightness of our bound, we perform numerical experiments to determine the highest learning rate possible without resulting in complete collapse. The results are presented in Figure 3, where we apply gradient descent to minimize the cosine similarity between each vector on a 100-dimensional unit ball. Specifically, let the class embedding matrix be denoted as $\mathbf{X} := [\mathbf{x}_1, \ldots, \mathbf{x}_k] \in \mathbb{R}^{100 \times k}$. Assuming the model has successfully collapsed the samples from each class into a single class embedding, the InfoNCE loss (Eq. (1)) can be simplified to:

$$\mathcal{L}_{class}(\mathbf{X}) := -\sum_{i \neq j}[\mathbf{X}^\top \mathbf{X}]_{ij}. \tag{3}$$

The left and middle figures of Figure 3 show the singular value spectrum of the same 30 class embeddings, with learning rates of 2.0 and 2.1, respectively. It is important to note that, according to Theorem 1, the upper bound for non-increasing class embedding means is 2.03. The left figure demonstrates successful learning with a low final loss, while the middle figure—where the learning rate is just 0.1 higher—results in complete collapse, as the singular values are nearly zero for all ranks except the first. Notably, collapse occurs early in training in the middle figure, aligning with the expectation from Figure 2, which suggests that overshooting may occur when the of Figure 3 are well-distributed across the space. Additionally, although the left figure successfully minimizes the total cosine similarity, there are signs of dimensional collapse, as the singular value of the last dimension gradually drops to zero. This indicates that adjusting the learning rate alone cannot prevent dimensional collapse, a topic we will explore further in Section 3.2.

In the right figure of Figure 3, we plot the maximum learning rates that prevent collapse over 5 consecutive trials for varying number of class embeddings, comparing them to the theoretical upper bound from Theorem 1. The theoretical upper bound closely aligns with the highest successful learning rate recorded. The reason our bound is tighter than the highest recorded successful learning rate is that we guarantee the class embedding mean is non-increasing over gradient steps (i.e., $\varepsilon = 0$), providing the safest bound for the learning rate.

## 3.2 Cosine-Similarity Causes Dimensional Collapse

There are three widely discussed methods for arranging $k$ class embeddings in an $m$-dimensional space, where $m \geq k$, to achieve optimal spatial utility. The first method is the Simplex Equiangular Tight Frame (ETF), as introduced in Zhu et al. (2021); Graf et al. (2021). This approach arranges the vectors on a hyperplane such that all vectors are equidistant from each other. ETF is frequently used to explain the phenomenon of dimensional collapse observed in the final layer of a neural network. The second method involves arranging $k$ vectors to be mutually orthogonal. The third method divides the $k$ vectors into equal-sized groups, where within each group, the vectors form pairs that point in opposite directions. Additionally, the vectors from different groups are orthogonal to each other.

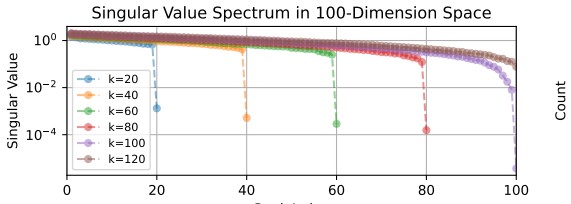 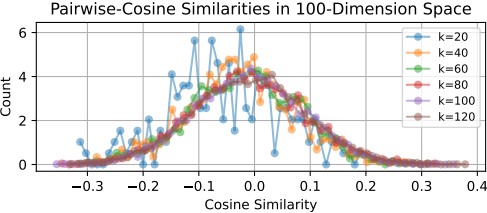

Figure 4: Numerical experiments illustrating dimensional collapse with $k$ class embeddings. The results show that minimizing total cosine similarity via gradient descent leads to convergence within a subspace of rank $k-1$ (**left**), while failing to preserve equal distances between the vectors (**right**).

While the second *"orthogonal"* method yields the most distinguishable classes from a linear algebra standpoint, the first *"simplex ETF"* and third *"inverted groups"* methods achieve the optimal value of the InfoNCE-equivalent $\mathcal{L}_{\text{class}}$, given as $-k$. This result is obtained under the same conditions outlined in Section 3.1, where the model effectively consolidates samples from the same class into a unified class embedding. However, none of these configurations are commonly observed in contrastive learning via gradient descent. Instead, a more common outcome is that the initially full-rank embedding space gradually collapses into a lower-dimensional subspace, where the embeddings are no longer equidistant. In this section, we take a microscopic perspective to explore how individual embeddings adjust to minimize total cosine similarity.

This observation is validated by a simple numerical experiment, where we minimize the pairwise cosine similarity $\mathcal{L}_{\text{class}}$ among all class embeddings using gradient descent. The class embeddings are randomly initialized from a Gaussian distribution and normalized to unit norm at each iteration. As shown in Figure 4, the results demonstrate that gradient descent consistently converges to a subspace of rank $k-1$, where the pairwise similarities vary significantly.

Building on this observation, we present Lemma 2 to illustrate that, in the case of full rank, the optimal movement for each individual class embedding is to align itself within the subspace spanned by all other class embeddings. Consequently, the class embedding closest to the others is most likely to be pushed into this subspace, eventually reaching a local optimum by forming a zero-mean subspace of rank $k-1$.

---

**Lemma 2.** *Let $\mathbf{X} = \{\mathbf{x}_1, \mathbf{x}_2, \ldots, \mathbf{x}_{k-1}\}$ be a set of $k-1$ linearly-independent unit-norm class embeddings in an $m$-dimensional space, where $m \geq k$. The optimal arrangement of the additional individual class embedding $\mathbf{x}_k$ that minimize the total cosine similarity score $\mathcal{L}_{class}$ (Eq. (3)) will result in $\mathbf{x}_k$ being linearly dependent on the remaining class embeddings in $\mathbf{X}$.*

---

Lemma 2 can be proven by constructing two matrices, $\mathbf{X}$ and $\mathbf{X}'$, each consisting of $k$ vectors. One matrix is full rank, while the other has rank $k-1$. The only difference between $\mathbf{X}$ and $\mathbf{X}'$ is that the vector $\mathbf{x}_k$ in $\mathbf{X}$ lies in a distinct dimension, whereas $\mathbf{x}'_k$ in $\mathbf{X}'$ is the normalized projection of $\mathbf{x}_k$ onto the subspace spanned by the remaining vectors in both $\mathbf{X}$ and $\mathbf{X}'$. It can be shown that $\mathcal{L}_{\text{class}}(\mathbf{X}) > \mathcal{L}_{\text{class}}(\mathbf{X}')$. The complete proof is provided in Appendix E.

## 4 OUR MODEL: CONTRASTIVE LEARNING WITH ORTHONORMAL PROTOTYPES (CLOP)

To avoid the issue of complete collapse and dimensional collapse, we introduce a novel approach (CLOP) that promotes point isolation by adding an additional term to the loss function for contrastive learning. Specifically, we initialize a group of *orthonormal prototypes*, serving as the target for each class, following the same idea of class embeddings. The number of orthonormal prototypes matches the total number of classes in the dataset. We then maximize the similarity between the orthonormal prototypes and the labeled samples in the training set.

Formally, let $\mathcal{S}$ be the labeled training set containing pairs of embeddings and labels, denoted as $\mathcal{S} = \{(\mathbf{z}_i, y_i) \mid i \in \{1, \ldots, |\mathcal{S}|\}\}$. The set of prototypes, denoted as $\mathcal{C}$, is defined as $\mathcal{C} = \{\mathbf{c}_1, \ldots, \mathbf{c}_k\}$, where $k$ represents the number of classes in the dataset. To generate the prototypes $\mathcal{C}$, we randomly

sample $k$ i.i.d. vectors from an $m'$-dimensional space, where $|\mathbf{z}_i| = m'$. Subsequently, we apply singular value decomposition (SVD) to obtain the orthonormal basis, denoted as $\mathcal{C}$. This ensures that each prototype $\mathbf{c}_i$ is initialized as a unit vector, orthogonal to all other prototypes, at the beginning of the training process. The CLOP loss is formulated as follows:

$$\mathcal{L}_{\text{CLOP}} = \mathcal{L}_{CL} + \lambda \sum_{i=1}^{|\mathcal{S}|} (1 - s(\mathbf{z}_i, \mathbf{c}_{y_i})), \tag{4}$$

where $\mathcal{L}_{CL}$ represents the primary contrastive learning loss (e.g., InfoNCE, DCL, SupCon), and $s(\cdot, \cdot)$ denotes the similarity metric, typically chosen to be the same metric used in $\mathcal{L}_{CL}$.

The primary objective of the CLOP loss is to align all embeddings corresponding to the same class towards a common target prototype, $\mathbf{c}_{y_i}$. Beyond the "gravitational force" and "repulsive force" provided by the main contrastive loss, the CLOP loss introduces a supervised "pulling force" that prevents collapse by isolating labeled embeddings into their own dimensions. It is important to note that, without additional constraints, samples outside of set $\mathcal{S}$ may still converge to other unspecified embeddings, potentially collapsing into a rank-1 subspace. However, a fundamental assumption in contrastive learning is that augmented samples are treated as being drawn from the same distribution as the original input data from the same class. Thus, the "gravitational force" between embeddings of the same class should pull unsupervised embeddings toward the target *class embedding* prototypes.

To illustrate the effectiveness of CLOP in mitigating embedding collapse, we conduct a straightforward experiment using synthetically generated data. We begin by initializing 500 input samples in a 3-dimensional space, categorized into three distinct classes. Samples within each class are initialized within the same linear subspace, with random Gaussian noise (mean 0, variance 0.05) added. A 3-layer Feedforward Neural Network (FFN) is then trained following SimCLR framework. For the baseline methods (InfoNCE, DCL, BarlowTwin, VICreg), the model is first trained using a self-supervised approach, where data augmentation is performed by adding random Gaussian noise (mean 0, variance 0.05) and randomly inverting the sign of the samples with a probability of 0.5. Following this training phase, the 10% labeled samples are used to train a K-Nearest Neighbors (KNN) classifier (with $k = 5$), simulating the fine-tuning phase, to predict the labels of the remaining unlabeled samples. For methods employing CLOP, the same 10% labeled samples are used for the CLOP loss during the initial training phase.

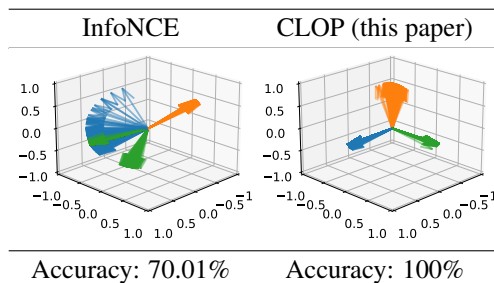

Figure 5: Impact of avoiding dimensional collapse with CLOP (proposed method) on InfoNCE for contrastive learning. A 3-layer FFN is trained on synthetic data with 10% labeled samples, and the output embeddings are visualized in 3D. KNN classification accuracy ($k = 5$) is reported, where the model is trained on 10% labeled data and tested on the remaining unlabeled data.

The output embeddings and KNN accuracy are partially presented in Table 5 and fully detailed in Table 5 within Appendix B. The color of each point in the output embedding visualizations corresponds to its ground truth label. As discussed in previous sections, methods utilizing sample-wise cosine similarity (e.g., InfoNCE and DCL) are expected to push the embeddings into a lower-dimensional subspace. This effect is clearly visible for InfoNCE in Table 5 and DCL in Table 5. Consistent with CLOP's goal of addressing dimensional collapse, we observe that the embeddings trained with CLOP show more distinct boundaries between classes, with each class being more orthogonal to the others. The accuracy results further corroborate the improvement in embedding quality facilitated by CLOP. Additionally, even for methods like VICReg and Barlow Twins (Table 5), which CLOP is not specifically designed for, we observe better learning outcomes when CLOP is applied. While the accuracy improvement for VICReg is marginal, the embedding visualizations clearly demonstrate that CLOP helps distribute the embeddings more evenly, potentially enhancing the model's ability to generalize to future tasks.

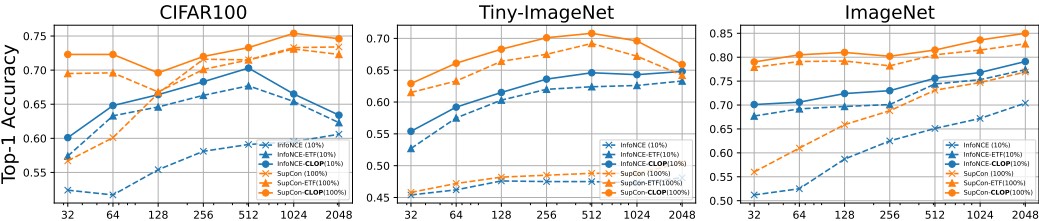

Figure 6: Top-1 classification accuracy across different batch sizes. The percentage of labels used for supervised training is indicated in the legend.

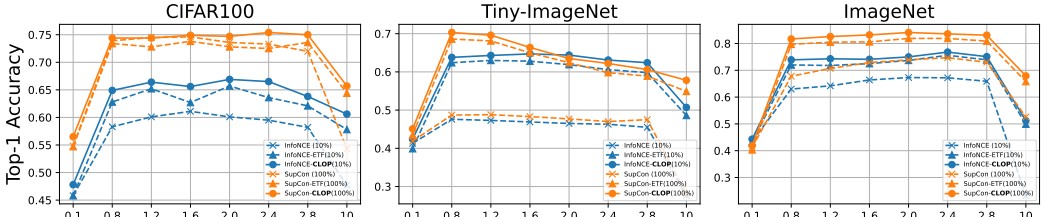

Figure 7: Top-1 classification accuracy across different different learning rates. The percentage of labels used for supervised training is indicated in the legend.

## 5 EXPERIMENT

In this section, we present the experimental results for image classification, conducted with various batch sizes and learning rates on the CIFAR-100 Krizhevsky et al. (2009), Tiny-ImageNet Le & Yang (2015), and full ImageNet Deng et al. (2009) datasets. For baseline methods, we implement the InfoNCE Wu et al. (2018) with a supervised linear classifier for semi-supervised learning and the SupCon Khosla et al. (2020) for fully-supervised learning. All experiments are performed using the SimCLR Chen et al. (2020a) framework with ResNet-50 He et al. (2016). In addition to these baselines, we introduce our novel loss function, CLOP, which incorporates a hand-tuned hyperparameter, $\lambda = 1$, as defined in the formulation (see Eq. (4)). To ensure a fair comparison with ETF, we also evaluate performance using ETF as prototypes instead of an orthonormal basis. For fully-supervised learning, we utilize all labels in the training datasets for both SupCon and CLOP. In the semi-supervised setting, we employ 10% of the labeled data for both linear classifier and CLOP training. For all experiments, we report both top-1 (Figure 6, 7) and top-5 (Appendix B) classification accuracy using the supervised linear classifier.

**CLOP Enables Smaller Batch Sizes.** We trained models with batch sizes of 32, 64, 128, 256, 512, 1024, and 2048 on CIFAR-100 and ImageNet for 200 epochs and on Tiny-ImageNet for 100 epochs. The learning rate was fixed at $(0.3 \times \text{batch size}/256)$ for optimal performance. The corresponding classification accuracies are presented in Figure 6. CLOP consistently outperformed the baseline methods across all batch sizes. As reported in the original papers Chen et al. (2020a); Khosla et al. (2020), contrastive learning performs optimally when the batch size exceeds 1024, a finding corroborated by our experiments. However, with the addition of CLOP, we observe significantly less performance degradation at smaller batch sizes. Remarkably, CLOP achieved similar accuracy with a batch size of 32 compared to the baseline SupCon with a batch size of 2048 for CIFAR-100.

**CLOP Prevents Collapse with Large Learning Rates.** We trained models with learning rates ranging from 0.1 to 10 on CIFAR-100 and ImageNet for 200 epochs and Tiny-ImageNet for 100 epochs, using a batch size of 1024. The corresponding classification accuracies are presented in Figure 7. Across both datasets, CLOP consistently outperforms the baseline methods. Moreover, as demonstrated by Theorem 1, excessively large learning rates can lead to complete collapse, as clearly observed in the baseline methods at a learning rate of 10 on both datasets. However, with the incorporation of CLOP into the loss function, we observe a significantly smaller performance degradation on both datasets.

**Ablation Study on $\lambda$ Tuning.** To evaluate the sensitivity of the tuning parameter $\lambda$ in CLOP, we trained the model with SupCon loss across different $\lambda$ values, keeping the batch size fixed at 1024. The classification accuracy on both CIFAR-100 and Tiny-ImageNet is reported in Table 1. We observe that the performance remains stable for $\lambda$ values ranging from 0.1 to 1.5, with $\lambda = 1.0$ and $\lambda = 1.5$ yielding the best overall performance.

| $\lambda$ | CIFAR-100 | | Tiny-ImageNet | |
|---|---|---|---|---|
| | Top-1 | Top-5 | Top-1 | Top-5 |
| 0.1 | 0.745 | 0.935 | 0.616 | 0.868 |
| 0.5 | 0.740 | 0.931 | 0.695 | **0.909** |
| 1.0 | 0.754 | **0.938** | **0.696** | 0.900 |
| 1.5 | **0.760** | 0.937 | **0.696** | 0.893 |

Table 1: Accuracy of different $\lambda$ values.

**Ablation Study on the Choice of Similarity Metric.** To evaluate the impact of different similarity functions on Eq. (4), we trained the same ResNet-50 architecture on CIFAR-100 using cosine similarity, Euclidean similarity, and Manhattan similarity. The results, presented in Table 5, indicate that cosine similarity, which aligns with $\mathcal{L}_{CL}$ in Eq. (4), achieves the highest performance.

| Similarity Metric | Top-1 | Top-5 |
|---|---|---|
| Cosine | **0.754** | **0.938** |
| Euclidean | 0.749 | 0.933 |
| Manhattan | 0.715 | 0.899 |

Table 2: Accuracy of different similarity metric.

**Ablation Study on the Choice of Augmentation** To evaluate the impact of augmentation strategies on CLOP, we trained the same ResNet-50 model on Tiny-ImageNet with a batch size of 1024. We selected three commonly used augmentation methods: 1) RandAugment: Augmentation with three operations randomly chosen from all image processing functions in PyTorch (e.g., padding, resizing, cropping, rotation, color jitter, Gaussian blur, inversion, contrast adjustment, equalization); 2) AutoAugment using the ImageNet policy proposed in Cubuk et al. (2018); 3) SimCLR Augmentation Policy.

| Augmentation | Top-1 | Top-5 |
|---|---|---|
| RandAug | **0.696** | **0.9** |
| AutoAug-Imagenet | 0.546 | 0.776 |
| SimCLR | 0.499 | 0.77 |

Table 3: Accuracy of different augmentation strategies.

## 6 CONCLUSION

In this paper, we conducted a comprehensive study on dimensional collapse in contrastive learning. Our contributions are threefold. First, we derived a theoretical upper bound on the learning rate, which prevents the embedding mean from shifting towards the boundary of the embedding space, ultimately avoiding complete collapse. Second, we identified the connection between dimensional collapse and cosine similarity by explaining the tendency of embeddings to reside on a hyperplane rather than occupying the full embedding space. To avoid dimensional collapse, we proposed a novel semi-supervised loss function, CLOP, which promotes better separation of the embedding space by pulling a subset of labeled training data towards orthonormal prototypes.

Our experiments on CIFAR-100, Tiny-ImageNet, and ImageNet demonstrated the effectiveness of CLOP, showing significant improvements in stability across varying learning rates and batch sizes. Additionally, our results indicate that CLOP enables the model to perform exceptionally well even with small batch sizes (e.g., 32), making the method particularly suitable for edge devices with limited memory.

In future work, we aim to further explore the use of pseudo-labeling for self-supervised learning with CLOP, reducing dependence on labeled data and extending the method's applicability to a broader range of contrastive learning tasks.

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

## A    RELATED WORK

| Method | Affinity Metric | Aff. to Prob. | Divergence Function | Top-1 | Top-5 |
|---|---|---|---|---|---|
| CPC-v2 Henaff (2020) | Cosine | Softmax | CrossEntropy | 71.5 | 90.1 |
| MOCO-v2 Chen et al. (2020b) | Cosine | Softmax | CrossEntropy | 71.1 | - |
| SimCLR Chen et al. (2020a) | Cosine | Softmax | CrossEntropy | 69.3 | 89.0 |
| Inclusion/Removal of Terms within InfoNCE | | | | | |
| PCL Cui et al. (2021) | Cosine | Softmax | CrossEntropy | 67.6 | - |
| LOOC Xiao et al. (2020) | Cosine | Softmax | CrossEntropy Variant | - | - |
| DCL Yeh et al. (2022) | Cosine | Decoupled Sftmx | CrossEntropy | 68.2 | - |
| RC Wang et al. (2022a) | Cosine | Softmax | CrossEntropy + L2 | 61.6 | - |
| Adjustments to the Similarity Function of InfoNCE | | | | | |
| Debiased Chuang et al. (2020) | Floor Cosine | Softmax | CrossEntropy | - | - |
| GCL Koishekenov et al. (2023) | Arccosine | Softmax | CrossEntropy | - | - |
| HCL Ge et al. (2023) | Cos. + Poincaré | Softmax | CrossEntropy | 58.5 | - |
| Innovations | | | | | |
| InfoMin Tian et al. (2020b) | Cosine | Softmax | MinMax CrossEntropy | **73.0** | 91.1 |
| VICref Bardes et al. (2021) | Euclidean | NA | Distance + Var + Cov | **73.1** | 91.1 |
| BT Zbontar et al. (2021) | Dimensional Cos. | NA | L2 | **73.2** | 91.0 |

Table 4: Overview of Novel Loss Functions and Baseline Results from 2020: Image Classification Accuracy on ImageNet1K with Unsupervised Learning and Full Label Fine-Tuning. The accuracy measurements are based on training a standard ResNet-50 with 24M parameters. The symbol '-' indicates that the corresponding metric was not reported in the original paper.

## B  EXTRA EXPERIMENTS ON CLOP

| Loss Fn. | InfoNCE | DCL | VICreg | BarlowTwins |
|---|---|---|---|---|
| Output Embeddings | | | | |
| Accuracy | 68.91% | 88.56% | 98.64% | 82.50% |
| Output Embeddings (w/ **CLOP**) | | | | |
| Accuracy (w/ **CLOP**) | 99.89% | 100% | 99.96% | 100% |

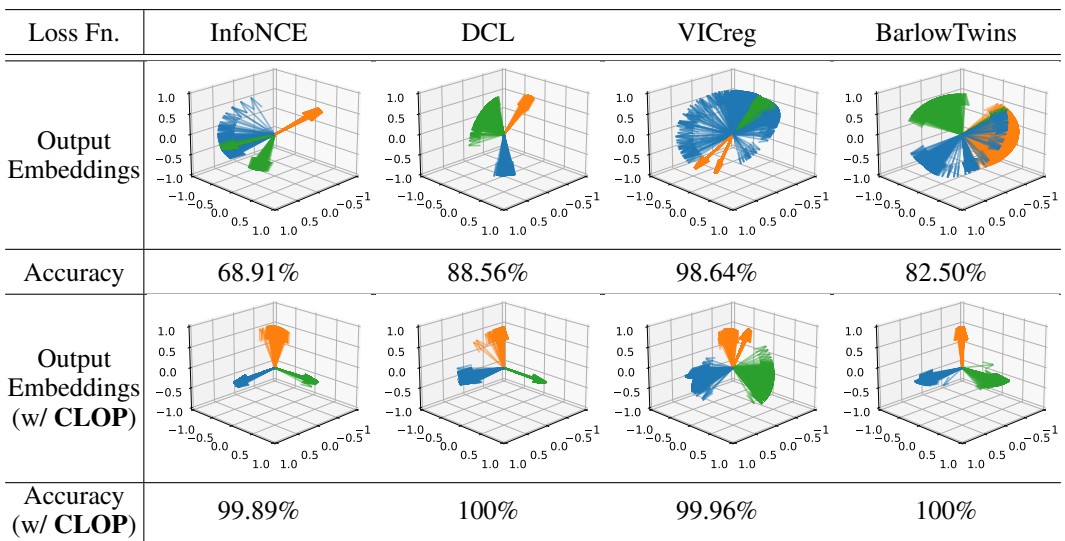

Table 5: Evaluation of Semi-Supervised Contrastive Learning on Synthetic Data with Different Loss Functions. A 3-layer Feedforward Neural Network (FFN) is trained on synthetic data with 10% labeled samples. Both the input and output embeddings reside in a 3-dimensional space, with the output embeddings visualized. The color of each point represents its ground truth label. Additionally, the K-Nearest Neighbors (KNN) classification accuracy ($k = 5$) is reported, where the model is trained on the labeled 10% of data and tested on the remaining unlabeled data.

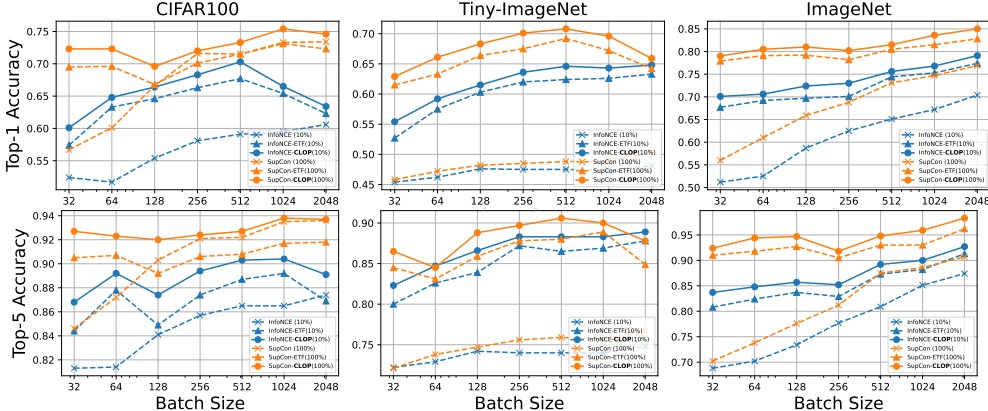

Figure 8: Top-1 classification accuracy across different batch sizes. The percentage of labels used for supervised training is indicated in the legend.

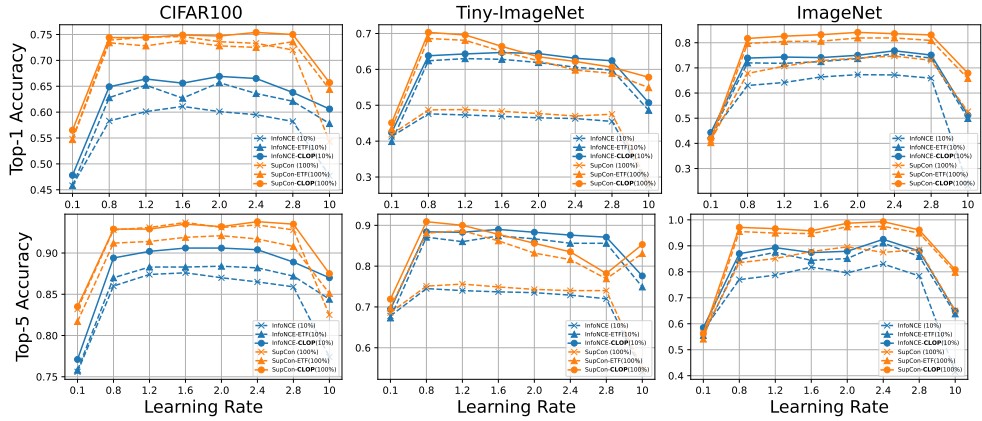

Figure 9: Top-1 classification accuracy across different different learning rates. The percentage of labels used for supervised training is indicated in the legend.

# C Proof of Theorem 1

*Proof of Theorem 1.* Consider $\mathcal{L}_i$ as the i-th loss term of $\mathcal{L}_{\text{InfoNCE}}$, defined by the following expression:

$$\mathcal{L}_i := -\log \mathbb{P}_i$$

where $\mathbb{P}_i$ denotes the probability that i-th embedding choose its positive pair as closest neighbor:

$$\mathbb{P}_i := \frac{\exp(\mathbf{z}_i^\top \mathbf{z}_{j(i)}/\tau)}{\exp(\mathbf{z}_i^\top \mathbf{z}_{j(i)}/\tau) + \sum_{a \notin \{i,j(i)\}} \exp(\mathbf{z}_i^\top \mathbf{z}_a/\tau)}$$

As detailed in Yeh et al. (2022), the gradient of $\mathcal{L}_i$ with respect to $\mathbf{z}_i$, $\mathbf{z}_{j(i)}$, and $\mathbf{z}_a$ can be derived as follows:

$$-\frac{\partial \mathcal{L}_i}{\partial \mathbf{z}_i} := (1 - \mathbb{P}_i)/\tau \left( \mathbf{z}_{j(i)} - \sum_{a \notin \{i,j(i)\}} \frac{\exp(\mathbf{z}_i^\top \mathbf{z}_a/\tau)}{\sum_{b \notin \{i,j(i)\}} \exp(\mathbf{z}_i^\top \mathbf{z}_b/\tau)} \mathbf{z}_a \right)$$

$$-\frac{\partial \mathcal{L}_i}{\partial \mathbf{z}_{j(i)}} := \frac{(1 - \mathbb{P}_i)}{\tau} \mathbf{z}_i$$

$$-\frac{\partial \mathcal{L}_i}{\partial \mathbf{z}_a} := -\frac{(1 - \mathbb{P}_i)}{\tau} \frac{\exp(\mathbf{z}_i^\top \mathbf{z}_a/\tau)}{\sum_{b \notin \{i,j(i)\}} \exp(\mathbf{z}_i^\top \mathbf{z}_b/\tau)} \mathbf{z}_i$$

In the standard setup of self-supervised learning, for any sample, there is one positive pair among $I$ and the remainder are all negative pairs. By aggregating all the gradient respect to a single sample, we have the gradient of InfoNCE respect to $\mathbf{z}_i$:

$$-\frac{\partial \mathcal{L}_{\text{InfoNCE}}}{\partial \mathbf{z}_i} := \frac{(1 - \mathbb{P}_i) + (1 - \mathbb{P}_{j(i)})}{\tau} \mathbf{z}_{j(i)} - \sum_{a \notin \{i,j(i)\}} \frac{(1 - \mathbb{P}_i)}{\tau} \frac{\exp(\mathbf{z}_i^\top \mathbf{z}_a/\tau)}{\sum_{b \notin \{i,j(i)\}} \exp(\mathbf{z}_i^\top \mathbf{z}_b/\tau)} \mathbf{z}_a$$

$$- \sum_{a \notin \{i,j(i)\}} \frac{(1 - \mathbb{P}_a)}{\tau} \frac{\exp(\mathbf{z}_i^\top \mathbf{z}_a/\tau)}{\sum_{b \notin \{a,j(a)\}} \exp(\mathbf{z}_a^\top \mathbf{z}_b/\tau)} \mathbf{z}_a$$

Now, considering the first scenario, where all embeddings equal, that means that $\mathbf{z}_i = \mathbf{z}_{j(i)} = \mathbf{z}_a = \mathbf{z}^*$ for all $a \in I$, the loss terms $\mathbb{P}_i$, $\mathbb{P}_{j(i)}$, and $\mathbb{P}_a$ converge to a constant $\mathbb{P}^*$, given by:

$$\mathbb{P}_i = \mathbb{P}_{j(i)} = \mathbb{P}_a = -\log \frac{1}{|I| - 1} := \mathbb{P}^*$$

Consequently, the gradient of $\mathcal{L}_{\text{InfoNCE}}$ with respect to $\mathbf{z}_i$ under this assumption reduces to zero, aligning with our expectations:

$$-\frac{\partial \mathcal{L}_{\text{InfoNCE}}}{\partial \mathbf{z}_i} = \frac{2(1 - \mathbb{P}^*)}{\tau} \mathbf{z}^* - 2(|I| - 2) \frac{(1 - \mathbb{P}^*)}{\tau} \frac{1}{|I| - 2} \mathbf{z}^* = 0$$

We establish the existence of local minima in scenarios where all embeddings are identical. Now, we consider the second scenario where all embeddings generated reside within the same rank-1 subspace. Denoting $\mathbf{z}^*$ as their unit basis, we can represent each embedding $\mathbf{z}_i$ as:

$$\mathbf{z}_i = \alpha \mathbf{z}^*, \quad \alpha \in \{-1, 1\}, \quad \forall i$$

The gradient of the loss function $\mathcal{L}_{\text{InfoNCE}}$ with respect to $\mathbf{z}_i$ simplifies to:

$$-\frac{\partial \mathcal{L}}{\partial \mathbf{z}_i} = \beta \mathbf{z}_i$$

Here, $\beta$ is a scalar that aggregates contributions from all relevant weights.

It is important to note that $\mathbf{z}^i$ represents the normalized output of the function $f$, with $\tilde{\mathbf{z}}_i$ denoting the original, unnormalized embedding. This implies the following relation:

$$-\frac{\partial \mathcal{L}}{\partial \tilde{\mathbf{z}}_i} = -\frac{\partial \mathcal{L}}{\partial \mathbf{z}_i} \frac{\partial \mathbf{z}_i}{\partial \tilde{\mathbf{z}}_i} = \frac{1}{\|\mathbf{z}_i\|_2} \left( \mathbb{I} - \frac{\mathbf{z}_i \mathbf{z}_i^\top}{\mathbf{z}_i^\top \mathbf{z}_i} \right) \beta \mathbf{z}_i = 0,$$

where $\mathbb{I}$ represents the identity matrix.

$\square$

# D PROOF OF THEOREM 1

*Proof.* At each step of gradient descent, every point $\mathbf{x}_i$ moves toward the negative of the mean of the other points with a step size $\eta$. Let the mean of all $k$ points before the gradient descent step be $\boldsymbol{\mu}^{(0)} := \frac{1}{k} \sum_{i=1}^{k} \mathbf{x}_i^{(0)}$. The update rule for the $i$-th point is given by:

$$\mathbf{x}_i^{(1)} = \mathbf{x}_i^{(0)} - \eta \frac{1}{k-1} \left( n \boldsymbol{\mu}^{(0)} - \mathbf{x}_i^{(0)} \right) = \left( 1 + \frac{\eta}{k-1} \right) \mathbf{x}_i^{(0)} - \eta \frac{k}{k-1} \boldsymbol{\mu}^{(0)}.$$

After the update, each point $\mathbf{x}_i^{(1)}$ is normalized to have unit norm, i.e., $\hat{\mathbf{x}}_i^{(1)} = \frac{\mathbf{x}_i^{(1)}}{\|\mathbf{x}_i^{(1)}\|}$. The expected norm of any updated vector is calculated as:

$$\mathbb{E}[\|\mathbf{x}_i^{(1)}\|^2] = \left( 1 + \frac{\eta}{k-1} \right)^2 \mathbb{E}\left[ \|\mathbf{x}_i^{(0)}\|^2 \right] + \eta^2 \left( \frac{k}{k-1} \right)^2 \mathbb{E}\left[ \|\boldsymbol{\mu}^{(0)}\|^2 \right]$$
$$- 2\eta \frac{k}{k-1} \left( 1 + \frac{\eta}{k-1} \right) \mathbb{E}\left[ \mathbf{x}_i^{(0)\top} \boldsymbol{\mu}^{(0)} \right]. \quad (5)$$

Since $\mathbf{x}_i^{(0)}$ is uniformly distributed on the surface of an $m$-dimensional unit ball, its covariance is $\mathrm{Cov}(\mathbf{x}_i^{(0)}) = \frac{1}{m} \mathbf{I}_m$. Therefore, the covariance of the mean is

$$\mathrm{Cov}(\boldsymbol{\mu}^{(0)}) = \mathrm{Cov}\left( \frac{1}{k} \sum_{i=1}^{k} \mathbf{x}_i^{(0)} \right) = \frac{1}{k^2} \sum_{i=1}^{k} \mathrm{Cov}(\mathbf{x}_i^{(0)}) = \frac{1}{km} \mathbf{I}_m.$$

The second moment of $\boldsymbol{\mu}^{(0)}$'s norm is the trace of its covariance matrix:

$$\mathbb{E}[\|\boldsymbol{\mu}^{(0)}\|^2] = \mathbb{E}[\boldsymbol{\mu}^{(0)\top} \boldsymbol{\mu}^{(0)}] = \mathrm{Tr}(\mathrm{Cov}(\boldsymbol{\mu}^{(0)})) = \frac{1}{km} \mathrm{Tr}(\mathbf{I}_m) = \frac{1}{k}. \quad (6)$$

Since $\mathbf{x}_i^{(0)}$ and $\mathbf{x}_j^{(0)}$ are independent for $i \neq j$, we know that $\mathbb{E}[\mathbf{x}_i^{(0)\top} \mathbf{x}_j^{(0)}] = 0$. Therefore,

$$\mathbb{E}[\mathbf{x}_i^{(0)\top} \boldsymbol{\mu}^{(0)}] = \mathbb{E}\left[ \frac{1}{k} \|\mathbf{x}_i^{(0)}\|^2 + \frac{1}{k} \sum_{j \neq i} \mathbf{x}_i^{(0)\top} \mathbf{x}_j^{(0)} \right] = \frac{1}{k} \mathbb{E}[\|\mathbf{x}_i^{(0)}\|^2]. \quad (7)$$

Since $\mathbb{E}[\|\mathbf{x}_i^{(0)}\|^2] = 1$, substituting equations Eq. (6) and Eq. (7) into Eq. (5), we obtain

$$\mathbb{E}[\|\mathbf{x}_i^{(1)}\|^2] = \left( 1 + \frac{\eta}{k-1} \right)^2 + \frac{k\eta^2}{(k-1)^2} - 2\eta \frac{k}{k-1} \left( 1 + \frac{\eta}{k-1} \right) \frac{1}{k}$$
$$= 1 + \frac{\eta}{k-1} - \frac{2\eta}{k} - 2\frac{\eta^2}{k(k-1)} + \frac{\eta^2 k}{(k-1)^2}.$$

Thus, the upper bound of the first-order expectation $\mathbb{E}[\|\mathbf{x}_i^{(1)}\|]$ can be denoted by $B$, where:

$$\mathbb{E}[\|\mathbf{x}_i^{(1)}\|] \leq \sqrt{1 + \frac{\eta}{k-1} - \frac{2\eta}{k} - 2\frac{\eta^2}{k(k-1)} + \frac{\eta^2 k}{(k-1)^2}} := B.$$

After the gradient descent step, the expectation of the new mean $\boldsymbol{\mu}^{(1)}$ is bounded as follows:

$$\boldsymbol{\mu}^{(1)} = \frac{1}{k} \sum_{i=1}^{n} \hat{\mathbf{x}}_i^{(1)} \geq \frac{1}{kB} \sum_{i=1}^{n} \mathbf{x}_i^{(1)} = \frac{1-\eta}{B} \boldsymbol{\mu}^{(0)}.$$

To prevent complete collapse in $\hat{\mathbf{X}}^{(1)}$, the mean should not increase by more than $1 + \varepsilon$, where $\varepsilon$ controls the tolerance for mean shift. By setting $\varepsilon = 0$, we can ensure that $\boldsymbol{\mu}^{(1)}$ does not exceed $\boldsymbol{\mu}^{(0)}$, thereby providing the safest bound for the learning rate. This implies that the learning rate $\eta$ must satisfy $\frac{1-\eta}{B} \leq 1 + \varepsilon$. This gives the condition:

$$(1-\eta)^2 \leq \left( 1 + \frac{\eta}{k-1} - \frac{2\eta}{k} - 2\frac{\eta^2}{k(k-1)} + \frac{\eta^2 k}{(k-1)^2} \right) (1 + \varepsilon)^2.$$

By setting $\varepsilon = 0$, we can simplify further to obtain the following inequality:

$$\left(-2 - \frac{1}{k-1} + \frac{2}{k}\right) + \eta \left(1 + 2\frac{1}{k(k-1)} - \frac{k}{(k-1)^2}\right) \leq 0.$$

For $k > 2$, we have $1 + 2\frac{1}{k(k-1)} - \frac{k}{(k-1)^2} > 0$, leading to the bound:

$$\eta \leq \frac{2 + \frac{1}{k-1} - \frac{2}{k}}{1 + 2\frac{1}{k(k-1)} - \frac{k}{(k-1)^2}} = \frac{2k^3 - 5k^2 + 5k - 2}{k^3 - 3k^2 + 4k - 2} = 2 + O\left(\frac{1}{k}\right).$$

Since $k$ is an integer, this bound is effectively $O(1)$. $\qquad\square$

## E   PROOF OF THEOREM 2

*Proof.* Consider a list of $k$ *class embeddings* in $m$-dimensional space, denoted as $\mathbf{X} := [\mathbf{x}_1, \ldots, \mathbf{x}_k] \in \mathbb{R}^{m \times k}$, where $m \geq k$ and each *class embedding* has unit norm (i.e., $\|\mathbf{x}_i\|_2 = 1$ for all $i$). The loss function is defined as the sum of pairwise cosine similarities between the *class embeddings* (refer to Equation 3). We first assume that all *class embeddings* are linearly independent, implying that $\text{Rank}(\mathbf{X}) = k$. Our first goal is to show that there always exists another matrix $\mathbf{X}' \in \mathbb{R}^{m \times k}$ with $\text{Rank}(\mathbf{X}') = k - 1$, such that $\mathcal{L}(\mathbf{X}) > \mathcal{L}(\mathbf{X}')$.

To construct such a matrix $\mathbf{X}'$, we select a *class embedding* $\mathbf{x}_k$ such that $\sum_{i \neq k} \mathbf{x}_i \neq 0$. The existence of such a *class embedding* $\mathbf{x}_k$ can be easily established by contradiction. Suppose, for the sake of contradiction, that for all $k$, $\sum_{i \neq k} \mathbf{x}_i = 0$. This would imply that each $\mathbf{x}_i$ must be zero, i.e., $\mathbf{x}_i = 0$ for all $i$, which contradicts the assumption that the $\text{Rank}(\mathbf{X}) = k$. Hence, such a *class embedding* $\mathbf{x}_k$ must exist. Since the *class embeddings* are linearly independent, $\mathbf{x}_k$ can be decomposed as a weighted sum of two unit-norm vectors: one orthogonal to all other *class embeddings*, and one lying in the subspace spanned by the remaining *class embeddings*. Specifically, we write:

$$\mathbf{x}_k = \eta \mathbf{x}_k^\perp + \sqrt{1 - \eta^2} \mathbf{x}_k^\|, \quad 0 < \eta \leq 1,$$

where $\mathbf{x}_k^\perp$ is orthogonal to all other *class embeddings* and $\mathbf{x}_k^\|$ lies in the subspace spanned by the remaining *class embeddings*. The loss associated with the $k$-th *class embedding* is:

$$\mathcal{L}_k(\mathbf{X}) = \sum_{i \neq k} \mathbf{x}_i^\top \mathbf{x}_k = \sum_{i \neq k} \mathbf{x}_i^\top \left(\eta \mathbf{x}_k^\perp + \sqrt{1 - \eta^2} \mathbf{x}_k^\|\right).$$

Since $\mathbf{x}_i^\top \mathbf{x}_k^\perp = 0$ for all $i \neq k$, we have:

$$\mathcal{L}_k(\mathbf{X}) = \sqrt{1 - \eta^2} \sum_{i \neq k} \mathbf{x}_i^\top \mathbf{x}_k^\|.$$

Now, construct $\mathbf{X}'$ by replacing $\mathbf{x}_k$ with $\mathbf{x}_k^\|$. The corresponding loss function becomes:

$$\mathcal{L}_k(\mathbf{X}') = \sum_{i \neq k} \mathbf{x}_i^\top \mathbf{x}_k^\|.$$

It is important to note that we can always find $\sum_{i \neq k} \mathbf{x}_i^\top \mathbf{x}_k^\| < 0$. If this sum is not negative, we can simply invert the sign of $\mathbf{x}_k^\|$, ensuring the sum becomes negative. Since $0 < \eta \leq 1$, it follows that $\sqrt{1 - \eta^2} < 1$. Consequently, $\sqrt{1 - \eta^2} \sum_{i \neq k} \mathbf{x}_i^\top \mathbf{x}_k^\| > \sum_{i \neq k} \mathbf{x}_i^\top \mathbf{x}_k^\|$. Therefore, we have $\mathcal{L}(\mathbf{X}) > \mathcal{L}(\mathbf{X}')$, as required. $\qquad\square$

