# OpenReview forum: "Preventing Collapse in Contrastive Learning with Orthonormal Prototypes (CLOP)"
_ICLR.cc/2025/Conference — Submitted to ICLR 2025_

### Official Review · Reviewer_tW63 · 2024-10-30

**Soundness:** 2
**Presentation:** 3
**Contribution:** 2
**Rating:** 5
**Confidence:** 5

**Summary:**

The paper introduces a contrastive learning method called CLOP, which aims to address the issue of neural collapse by leveraging orthonormal prototypes. The authors theoretically analyze the collapse phenomenon in self-supervised and semi-supervised contrastive learning setups and propose a novel loss function to mitigate collapse by enforcing orthogonal subspace constraints. Experiments on CIFAR-100 and Tiny-ImageNet suggest that the proposed method stabilizes performance under varying batch sizes and learning rates.

**Strengths:**

*Theoretical Analysis:* The paper provides a theoretical analysis of neural collapse, focusing on the impact of large learning rates in contrastive learning settings. The theoretical insights could offer useful guidance for practitioners concerned with collapse issues in representation learning.

*New Loss Design:* The proposal of CLOP, a loss function to enforce orthogonality among embeddings, is a potentially valuable direction for addressing neural collapse. It could add diversity to the current strategies used in self-supervised and semi-supervised learning frameworks.

**Weaknesses:**

*Motivation:* The primary motivation of CLOP—to prevent neural collapse and enhance feature diversity—bears significant similarity to VICReg, which also aims to mitigate collapse by enforcing variance, invariance, and covariance constraints on representations. Although CLOP takes an orthogonal approach, it does not sufficiently differentiate itself from VICReg in either theoretical justification or practical application. The theoretical analysis mainly emphasizes the necessity for spatial separation, an objective already addressed by VICReg, leading to questions about the novelty of the approach.

*Insufficient Comparative Analysis:* While VICReg is briefly acknowledged, the paper lacks a robust empirical comparison with VICReg to establish CLOP’s superiority or complementary advantages. In particular, the experiments fail to demonstrate a distinct performance edge over VICReg, which diminishes the contribution of the proposed method. For this paper to make a convincing case for CLOP, direct comparisons, especially in scenarios prone to collapse, would be necessary.

*Experimental Design Issues:* The choice of CIFAR-100 and Tiny-ImageNet as benchmark datasets provides limited insights into the scalability and effectiveness of CLOP in larger and more complex real-world scenarios. Furthermore, while CLOP is tested under different batch sizes and learning rates, the experiments do not include variations in data augmentation or pretext tasks, which are known to impact the success of contrastive learning methods.

**Questions:**

Please see weaknesses

---

> ### Author Response · Authors · 2024-11-29
>
> We appreciate the reviewer’s constructive feedback and the opportunity to address their concerns. Below, we provide detailed responses to each point:
>
> 1. Motivation and Differentiation from VICReg:
>
> We thank the reviewer for highlighting concerns about the motivation and novelty of CLOP compared to VICReg. While VICReg mitigates collapse by enforcing variance, invariance, and covariance constraints in self-supervised learning scenarios, CLOP takes a fundamentally different approach by explicitly leveraging orthonormal prototypes within a semi-supervised learning framework to enforce spatial separation among embeddings. To address the reviewer’s concerns, we evaluated CLOP against VICReg on ImageNet, and the top-1 accuracy results are provided below. Additional results can be found in Figures 6 and 7.
>
> | Batch Size   | 32    | 64    | 128   | 256   | 512   | 1024  | 2048  |
> |--------------|-------|-------|-------|-------|-------|-------|-------|
> | VICReg       | 0.525 | 0.524 | 0.598 | 0.621 | 0.637 | 0.668 | 0.713 |
> | SupCon-CLOP   | 0.790 | 0.805 | 0.810 | 0.802 | 0.815 | 0.836 | 0.850 |
>
> | Learning Rate | 0.1   | 0.8   | 1.2   | 1.6   | 2.0   | 2.4   | 2.8   | 10.0  |
> |----------------|--------|--------|--------|--------|--------|--------|--------|--------|
> | VICReg         | 0.528  | 0.676  | 0.681  | 0.621  | 0.634  | 0.615  | 0.579  | 0.361  |
> | SupCon-CLOP     | 0.418  | 0.817  | 0.826  | 0.832  | 0.841  | 0.836  | 0.831  | 0.679  |
>
> 2. Limited Experiments on Small-Scale Datasets:
>
> We acknowledge the reviewer’s concern regarding the generalizability of our findings given the limited dataset scope. To address this, we have included additional experiments on the ImageNet-1K dataset in the revised manuscript (Figures 6 and 7).
>
> 3. Ablation Study on Data Augmentation:
>
> We appreciate the reviewer’s observation regarding the role of data augmentation in contrastive learning methods. In the revised version, we have added an ablation study to analyze the effect of various augmentation strategies, including Random Augmentation, AutoAugment, and SimCLR’s augmentation strategy, on CLOP’s performance (Table 3).
>
> We thank the reviewer again for their thoughtful feedback. We hope that these revisions satisfactorily address all concerns.

---

### Official Review · Reviewer_Wfg9 · 2024-11-01

**Soundness:** 1
**Presentation:** 2
**Contribution:** 2
**Rating:** 5
**Confidence:** 3

**Summary:**

This paper considers the problem of neural collapse in contrastive learning. The authors theoretically investigate the effect of large learning rate in contrastive learning with cosine similarity and provide a bound for the learning rate to avoid the collapse. The authors also propose a semi-supervised contrastive loss. Experimental results on small image datasets support the claim.

**Strengths:**

A theoretical upper bound of the learning rate for collapse is provided.

**Weaknesses:**

- The definition of neural collapse is incorrect. Neural collapse refers to the phenomenon that within-class variance becomes zero. Although it would result in low-rank representations, "embeddings converge into a lower-dimensional space" is insufficient to properly describe neural collapse. Please refer to [Papyan et al.] that mainly discuss neural collapse, which is not cited in this paper.

[Papyan et al.] Prevalence of Neural Collapse during the terminal phase of deep learning training. PNAS 2021.

- The bound seems to be not practical. 2 is too large learning rate for contrastive learning.

- While experimental results are sensitive to the batch size, there is no discussion about the batch size throughout the theoretical analysis.

- Experimental setting is limited to small image datasets without transfer learning, so its generalizability is questionable.

- No ablation study on the choice of the similarity function, while the authors claim that the proposed method is better than the one with consine similarity.

- Does Lemma 2 still hold when Eq. (3) is used?

**Questions:**

Please address concerns above.

---

> ### Author Response · Authors · 2024-11-29
>
> We appreciate the reviewer’s constructive feedback and the opportunity to address their concerns. Below, we respond to each point in detail:
>
> 1. Conflation of Dimensional Collapse and Neural Collapse:
>
> We thank the reviewer for highlighting this issue. In the revised version of the paper, we have corrected all references to appropriately use the term dimensional collapse.
>
> 2. Practicality of the Learning Rate Bound:
>
> We understand the reviewer’s concern that the proposed bound may not appear practical for common learning rates used in contrastive learning. Our theoretical bound is designed to provide insights into the relationship between learning rate and collapse under the assumption that all vectors are unit-norm, rather than prescribing exact values for real-world applications. To clarify this, we have revised the discussion to express the upper bound of the learning rate as a constant .
>
> 3. Lack of Discussion on Batch Size in Theoretical Analysis:
>
> We appreciate the reviewer’s observation regarding the lack of batch size considerations in the theoretical framework. In the theoretical section, our analysis is based on the number of classes k. However, in real-world learning scenarios, unlabeled samples are often treated as their own unique class, effectively making the batch size and the number of classes interchangeable in this context. We will clarify this relationship and ensure that this concern is addressed comprehensively in the final version of the paper.
>
> 4. Limited Experiments on Small-Scale Datasets:
>
> We acknowledge the reviewer’s concern regarding the generalizability of our findings given the limited dataset scope. In the revised manuscript, we have included additional experiments on the ImageNet-1K dataset (Figures 6 and 7).
>
> 5. No Ablation Study on Similarity Function:
>
> We agree with the reviewer that an ablation study on the choice of the similarity function is critical to substantiate our claims. In the revised manuscript, we have added an ablation study comparing the performance of various similarity functions, including cosine, Euclidean, and Manhattan distances, under the same experimental settings (Table 2). This analysis demonstrates the advantages of using cosine similarity in our proposed method and validates our claims.
>
> 6. Validity of Lemma 2 with Eq. (3):
>
> Yes, Lemma 2 is derived under the same setting as Section 3.1 and uses the loss function described in Eq. (3).
>
> We thank the reviewer again for their thoughtful feedback. We hope that these revisions address all concerns.

---

### Official Review · Reviewer_RdbF · 2024-11-03

**Soundness:** 2
**Presentation:** 3
**Contribution:** 2
**Rating:** 5
**Confidence:** 4

**Summary:**

This paper addresses the dimensional collapse problems within the semi-supervised contrastive learning paradigm. This paper first provides theoretical insights into the influence of large learning rates and cosine similarities on dimensional collapse. To mitigate this issue, the authors introduce a method, termed CLOP, aimed at enhancing subspace separation.  Experimental results on CIFAR and Tiny-ImageNet demonstrate the performance improvements of the algorithm.

**Strengths:**

1.	The paper tackles the critical issue of dimensional collapse within semi-supervised learning scenarios.
2.	This paper provides some theoretical clues to support their claims.
3.	The paper is generally well structured and easy to follow.

**Weaknesses:**

1.	The paper appears to conflate two distinct concepts: dimensional collapse and neural collapse. Dimensional collapse in semi-supervised learning (SSL) typically refers to the limited dimensionality of learned SSL features, while neural collapse is associated with a desirable state in supervised training marked by several good qualities such as intra-class alignment and inter-class separation (properties NC1-NC4).
2.	The claim regarding orthogonal structures achieving the most distinguishable classes relative to the Simplex ETF structure is unclear. Simplex ETF theoretically maximizes angular separation, reaching a pairwise cosine value of -1/(k-1), whereas orthogonal structures attain pairwise cosine values of zero.
3.	The issue of dimensional collapse is primarily addressed by the SSL community; however, the paper's focus is on semi-supervised methods, yet it lacks comparison with other semi-supervised baselines.
4.	The proposed method should be compared with ETF-based(or uniform variants) methods[1] to adequately demonstrate its effectiveness. Moreover, to comprehensively address dimensional collapse, additional self-supervised experiments are recommended, including comparisons with ETF-based SSL methods [2] and other methods targeting dimensional collapse [3,4].
5.	Experiments are limited to small-scale datasets, such as CIFAR and Tiny-ImageNet, which may restrict the generalizability of the findings.
6.	The performance improvements are not significant on larger batch sizes.

[1] Targeted supervised contrastive learning for long-tailed recognition.

[2] Combating Representation Learning Disparity with Geometric Harmonization.

[3] Understanding dimensional collapse in contrastive self-supervised learning.

[4] Variance-invariance-covariance regularization for self-supervised learning.

**Questions:**

please refer to the weakness part.

---

> ### Author Response · Authors · 2024-11-29
>
> We appreciate the reviewer’s constructive feedback and the opportunity to address their concerns. Below, we respond to each point raised in detail:
>
> 1. Conflation of Dimensional Collapse and Neural Collapse:
>
> We thank the reviewer for pointing out this issue. In the revised version of the paper, we have corrected all references to appropriately use the term dimensional collapse where applicable.
>
> 2. Clarity Regarding Orthogonal Structures vs. Simplex ETF:
>
> We appreciate the reviewer’s insightful comments regarding the Simplex ETF and its optimal cosine values. As discussed in Section 3.2, while the Simplex ETF theoretically achieves superior angular separation, we observed in practice that it is rarely achieved by merely minimizing the loss. Instead, the optimization often results in hyperplane-based formulations with zero-mean embeddings.
> Building on this observation, our proposed method, CLOP, encourages embeddings to align with different subspaces. This results in full-rank embeddings, rather than degenerate hyperplanes. To further substantiate our claims, we have added new experiments comparing the use of orthonormal structures versus Simplex ETF prototypes. These results are presented in Figures 6 and 7 of the revised paper.
>
> 3. Limited Experiments on Small-Scale Datasets:
>
> We acknowledge the reviewer’s concern about the generalizability of our findings given the limited dataset scope. In the revised version, we have included additional experiments on the ImageNet-1K dataset (Figures 6 and 7), addressing concerns about scalability and supporting the broader applicability of our approach.
>
> 4. Performance Improvements on Larger Batch Sizes:
>
> We acknowledge the reviewer’s observation regarding the performance improvements under larger batch sizes. Our aim is to demonstrate that under small batch sizes, our method performs relatively well compared to competing approaches with larger batch sizes, making it particularly suitable for scenarios constrained by computational memory resources. For larger batch sizes, the increased diversity of positive and negative pairs naturally reduces the risk of collapse, which lessens the relative impact of our method. Nonetheless, we observe that our approach remains competitive in these settings.
>
> We thank the reviewer again for their valuable feedback. We hope that the revisions adequately address the concerns raised.

---

### Meta-Review · Area_Chair_5awb · 2024-12-20

**Metareview:**

The paper addresses the problem of dimensional collapse in semi-supervised contrastive learning (SSL) by introducing a novel method termed CLOP. The authors provide a theoretical analysis of collapse in SSL, focusing on the effects of large learning rates and cosine similarities. CLOP is designed to enhance subspace separation using orthonormal constraints. The experimental results, primarily conducted on CIFAR-100 and Tiny-ImageNet, show some performance improvements, but significant concerns remain regarding the scope, novelty, and experimental rigor of the work.

The distinction between CLOP and existing methods, such as VICReg, is not sufficiently clear. Both approaches aim to mitigate collapse, but CLOP does not convincingly differentiate itself in terms of theoretical justification or practical utility. The conflation of dimensional collapse and neural collapse in the initial manuscript weakens the theoretical foundation, though this was partially addressed in the rebuttal.

The experiments are limited to small-scale datasets (CIFAR-100, Tiny-ImageNet) and do not convincingly demonstrate the method’s generalizability to larger datasets or other domains. While the rebuttal included results on ImageNet-1K, these additions do not fully address the concerns about scalability or robustness.

**Additional Comments On Reviewer Discussion:**

No reviewer gives a positive score.

---

### Decision · Program_Chairs · 2025-01-22

Reject